# Anisotropic fluid with phototunable dielectric permittivity

Hiroya Nishikawa [1✉], Koki Sano [1,2,3✉] & Fumito Araoka [1✉]

Dielectric permittivity, a measure of polarisability, is a fundamental parameter that dominates various physical phenomena and properties of materials. However, it remains a challenge to control the dielectric permittivity of materials reversibly over a large range. Herein, we report an anisotropic fluid with photoresponsive dielectric permittivity ($200 < \varepsilon < 18{,}000$) consisting of a fluorinated liquid-crystalline molecule (96 wt%) and an azobenzene-tethered phototrigger (4 wt%). The reversible *trans-cis* isomerisation of the phototrigger under blue and green light irradiation causes a switch between two liquid-crystalline phases that exhibit different dielectric permittivities, with a rapid response time ($<30$ s) and excellent reversibility ($\sim100$ cycles). This anisotropic fluid can be used as a flexible photovariable capacitor that, for example, allows the reversible modulation of the sound frequency over a wide range ($100 < f < 8500$ Hz) in a remote manner using blue and green wavelengths.

[1] RIKEN Center for Emergent Matter Science, 2-1 Hirosawa, Wako, Saitama 351-0198, Japan. [2] JST PRESTO, 4-1-8 Honcho, Kawaguchi, Saitama 332-0012, Japan. [3] Present address: Department of Chemistry and Materials, Faculty of Textile Science and Technology, Shinshu University, 3-15-1 Tokida, Ueda, Nagano 386-8567, Japan. ✉email: hiroya.nishikawa@riken.jp; koki.sano@riken.jp; fumito.araoka@riken.jp

Light is a fascinating tool for controlling material properties because of its non-contact and non-invasive nature, instantaneous and spatiotemporal controllability, and availability of various wavelengths[1,2]. Light irradiation often stimulates a phase transition of matter or a change in its electronic, magnetic, and/or quantum states, enabling the phototuning of the fundamental material properties. So far, various material properties, such as optical absorption[3] and reflection[4], mechanical elasticity[5], thermal[6] and electrical[7] conductivity, and magnetic permeability[8], have been controlled using the light stimulus[9–14]. However, it remains a significant challenge to photocontrol the dielectric permittivity of the material considerably in a reversible manner, although dielectric permittivity significantly affects the optical[15,16], mechanical[17], and electrical[18] properties, as well as molecular and colloidal interactions of materials[19]. Some crystals, such as barium titanate ($\varepsilon = {\sim}10{,}000$), exhibit large dielectric permittivity, but generally lack large and reversible tunability of their dielectric permittivity because of the difficulty in introducing stimuli-responsive units[20]. In sharp contrast, fluidic materials, including liquid crystals, can turn photoresponsive when doped with a phototrigger[4,21]. However, their dielectric permittivity is usually small ($\varepsilon < {\sim}100$) owing to the difficulty in maintaining a large polar order[22,23]. Therefore, if a fluidic material with a large dielectric permittivity is available, the incorporation of an appropriate phototrigger may result in a large and reversible phototunable dielectric permittivity.

In early 2017, Mandle et al. reported that a pear-shaped liquid crystal (LC) molecule (RM734) exhibits a new type of nematic (N) phase with unusual defect lines[24,25]; this LC phase was later named a splay nematic phase by Mertelj et al., owing to its periodic splay deformation perpendicular to the director[26–28]. During the same time, Nishikawa et al. reported that a fluorinated LC molecule bearing a 1,3-dioxane unit in the mesogenic core (DIO; Fig. 1a) adopts a highly polar nematic phase with gigantic dielectric permittivity ($\varepsilon > 10{,}000$)[29]. Recently, the splay nematic phase of RM734 and the highly polar nematic phase of DIO were assigned to a ferronematic ($N_F$) phase with a ferroelectric characteristic[30–33]. Importantly, DIO exhibits the N–$N_F$ phase transition across the additional mesophase (M) which is presumably having a local antiferroelectric nematic order[32,33] and the $N_F$–M phase transition is accompanied by a large change in dielectric permittivity ($N_F$ of DIO: $\varepsilon > 10{,}000$; M of DIO: $\varepsilon < 200$). Therefore, we speculated that the DIO system can be used to achieve a large and reversible change in its dielectric permittivity under light irradiation.

In the present study, we succeeded in photoswitching between the two phases (i.e. $N_F$ and M phases) of DIO by doping it with a newly designed azobenzene-tethered phototrigger (see Supplementary Figs. 1–6 for the characterisation data). As a result, the dielectric permittivity could be photocontrolled over a large range ($200 < \varepsilon < 18{,}000$) with a rapid response time ($<30$ s) and excellent reversibility (~100 cycles). Notably, the relative tunability of the dielectric permittivity of this system (99.5%) is the highest ever reported for any material, including organic and/or inorganic materials, with stimuli-responsive dielectric permittivity (see Supplementary Table 1). Furthermore, our new phototrigger based on a fluorinated azobenzene derivative undergoes slow thermal relaxation from the cis form to trans one ($t_{1/2} = 128$ days) and exhibits rapid and reversible trans-cis photoisomerisation under blue light (BL) and green light (GL) irradiation, a feature suitable for developing wearable and flexible devices. As a demonstration of its applicability, this fluidic material was used to develop a flexible photovariable capacitor that allows the reversible modulation of the sound frequency over a wide range ($100 < f < 8500$ Hz) in a remote manner. Such fluidic materials with a large and reversible phototunability of dielectric permittivity can be used as a solvent or constituent of soft materials and flexible devices.

## Results and discussion

**Characterisation of the ferronematogenic molecule (DIO) and phototrigger molecules.** The key material of the present work is the aforementioned DIO (Fig. 1a) that exhibits three mesophases during the cooling process [N (173.6–84.5 °C) → M (84.5–68.8 °C) → $N_F$ (68.8–34 °C)][29]. The dielectric permittivities of the N and M phases are approximate of the order of 100, whereas the $N_F$ phase exhibits an extremely large dielectric permittivity of >10,000. It is noteworthy that such a large dielectric permittivity does not depend on the macroscopic orientation of the DIO molecules (Supplementary Fig. 7 and Supplementary Note 1); therefore, we used non-oriented DIO molecules in this study. Usually, the local molecular order in LCs can be easily altered by incorporating a molecular dopant, thereby causing a phase transition[4,12–14,34,35]. Hence, we envisioned that incorporating an appropriate phototrigger into DIO might enable a photoinduced phase transition between the M and $N_F$ phases, allowing large tunability of its dielectric permittivity.

In this work, we employed three types of azobenzene-tethered phototrigger molecules that undergo reversible trans-cis photoisomerisation: Azo-H, Azo-Me and Azo-F (Fig. 1). Azo-H (also known as BMAB) is a common phototrigger that induces a phase transition between the nematic and isotropic phases of LCs; the phase transition occurs because of the disordering of the host LC molecules by the large volume changes stemming from trans to cis isomerisation[34,35]. Figure 1b, c shows the photoresponsive optical properties of Azo-H in a solution state, where two distinct UV/visible (vis) signals at 360 and 440 nm are due to the $\pi$–$\pi^*$ and $n$–$\pi^*$ transitions, respectively, of the trans- and cis-azobenzene isomers, respectively (see also Supplementary Fig. 8). Although this phototrigger facilitates a rapid and reversible interchange between the two states with alternate UV/vis irradiation, it has severe disadvantages: (i) the low thermal stability of the cis-isomer[36,37] and (ii) a narrow window of available wavelengths for photoisomerisation[38]. Owing to (i), the half-life of Azo-H is only 1.1 days, which is not suitable for practical applications (Fig. 1d). Such low thermal stability is generally caused by electronic repulsion between the nitrogen lone pairs in cis-azobenzene[36,37]. Further, when the absorption window is narrow (for the case (ii)), UV irradiation, which is essential for inducing the cis-trans isomerisation, sometimes leads to the degradation of Azo-H owing to the non-selective absorption of the high-energy UV light. With the use of Azo-Me[39], which possesses two methyl groups instead of two protons on the benzene ring of Azo-H, an almost identical photochemical reaction was observed (Fig. 1e, f and Supplementary Fig. 8). Importantly, the half-life of Azo-Me increased significantly to 65 days (Fig. 1g) because the cis isomer is structurally fixed by the intermolecularly locked methyl groups[39]. However, disadvantage (ii) still persists. Thus, in our work, to overcome the intrinsic disadvantages of the azobenzene unit, we designed and synthesised a new phototrigger molecule, a tetra-ortho-substituted azobenzene derivative (Azo-F; Supplementary Figs. 1–6). The $n$–$\pi^*$ bands of the cis- and trans-isomers of Azo-F were successfully separated (see also Supplementary Note 2), and they appeared at 420 and 450 nm, respectively, resulting in a wider operation range than those of Azo-H and Azo-Me. Indeed, we could use GL (525 nm) and BL (415 nm) to reversibly induce the photoisomerisation of Azo-F in a solution state (Fig. 1h, i and Supplementary Fig. 8). In addition, compared to Azo-H and Azo-Me, Azo-F exhibited an extremely long half-life of 128 days (Fig. 1j), which can be attributed to the electron-withdrawing

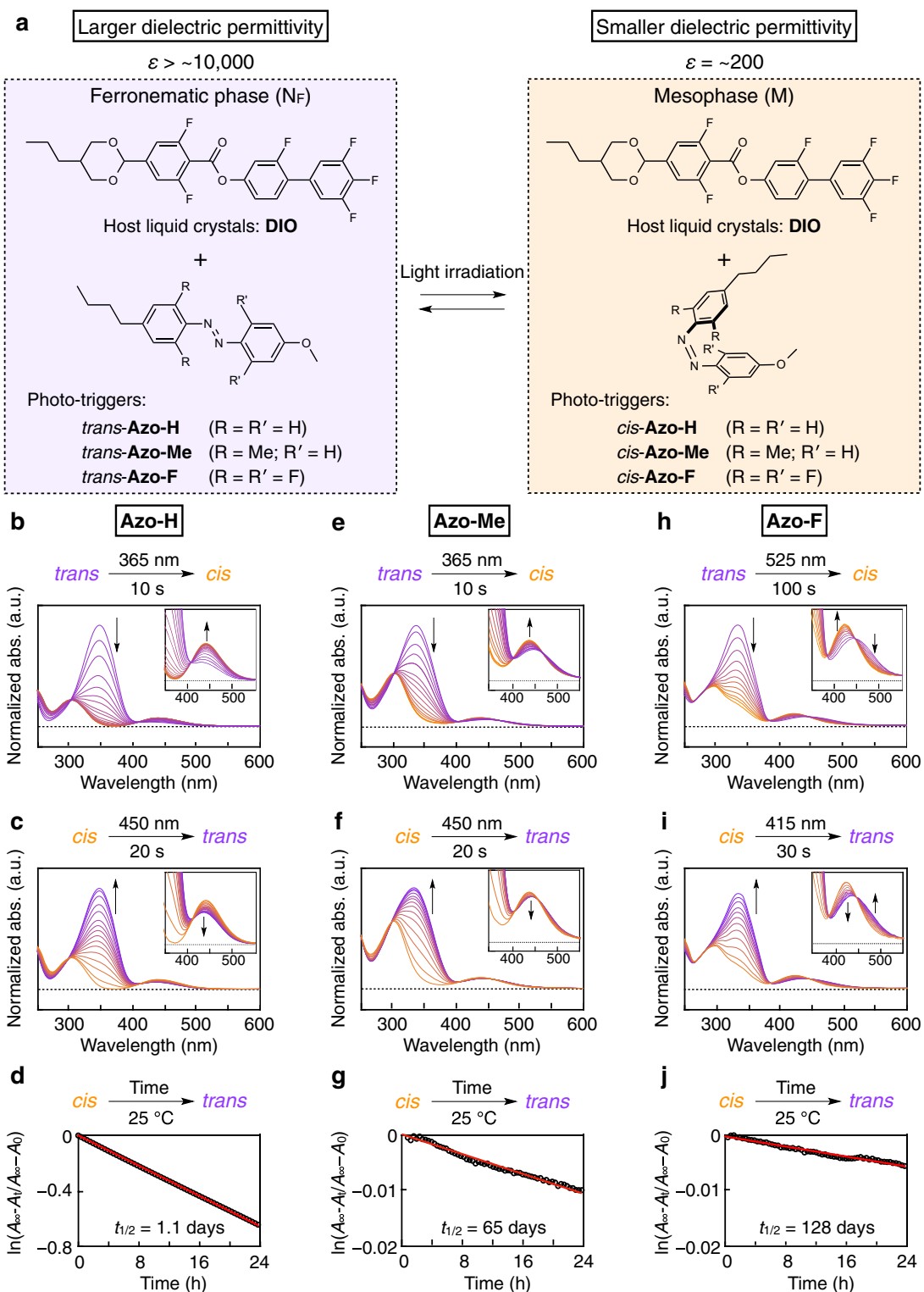

**Fig. 1 Phototunable system with gigantic dielectric properties. a** Schematic illustration of a liquid-crystalline (LC) blend consisting of a ferronematogen (DIO) and an azobenzene-tethered phototrigger (Azo-H, Azo-Me or Azo-F) with phototunable dielectric permittivity. **b**–**j** UV–visible spectra of Azo-H (**b**, **c**), Azo-Me (**e**, **f**) and Azo-F (**h**, **i**) in MeCN after light irradiation to induce their *cis* states [$\lambda_{ex}$ of Azo-H (**b**) and Azo-Me (**e**): 365 nm; $\lambda_{ex}$ of Azo-F (**h**): 525 nm] and after light irradiation to induce their *trans* states [$\lambda_{ex}$ of Azo-H (**c**) and Azo-Me (**f**): 450 nm; $\lambda_{ex}$ of Azo-F (**i**): 415 nm]. Time-dependent absorbance profiles of the phototrigger molecules (*cis* states in dimethyl sulfoxide; 25 °C in dark) under excitation at a fixed wavelength [Azo-H (**d**): 355 nm; Azo-Me (**g**) and Azo-F (**j**): 340 nm]. Red lines indicate linear fits [$R^2 = 1.00$, 0.99, and 0.98 for panels (**d**), (**g**) and (**j**), respectively].

fluorine atoms in the azobenzene core, which decreased the electron density and hence repulsive interaction[36,37]. Thus, we selected Azo-F as the best candidate for phototuning the dielectric permittivity of the LC.

**Photoinduced mesophase (M)–ferronematic (N$_F$) phase transition of LC blends**. Next, we incorporated Azo-F into the ferronematogen (DIO) to realise a photoresponsive LC blend. The UV/vis spectral changes of the LC blend ([Azo-F] = 4 wt%) upon alternate GL/BL irradiation confirmed the quick and reversible photoisomerisation behaviour, similar to that in the solution state (Supplementary Fig. 9). To investigate the effect of the doping concentration of Azo-F ([Azo-F]) in the ferronematogen (DIO), we performed dielectric relaxation measurements before and during GL irradiation (i.e. for *trans* and *cis* states). Figure 2a displays the temperature-dependence of the apparent dielectric permittivity ($\varepsilon'$) values of LC blends with different Azo-F contents (0–8 wt%) during a cooling process ($T = 70$–$30\,°C$). When pristine DIO ([Azo-F] = 0 wt%) was used, its dielectric permittivity was ~200 corresponding to the M phase ($T = 84$–$68\,°C$), and the value increased considerably to ~17,000 at ~68 °C and remained constant corresponding to the N$_F$ phase ($T < 68\,°C$). The LC blends ([Azo-F] = 1–8 wt%) in the *trans* state exhibited similar permittivity behaviours, except that the phase-transition temperature decreased gradually with increasing [Azo-F]. In addition, the polarised optical microscopic images of the LC blends ([Azo-F] = 1–8 wt%) were almost identical to those of pristine DIO (Supplementary Figs. 10 and 11 and Supplementary Note 3). The enthalpy of the M–N$_F$ phase transition ($\Delta H$) of the LC blends ([Azo-F] = 1–8 wt%) was also in good agreement with the $\Delta H$ calculated using the volume function of Azo-F (Supplementary Table 2 and Supplementary Note 3). These results confirm that the LC blend inherits the M–N$_F$ phase transition behaviour of pristine DIO. During GL irradiation ($I = 180\,\mathrm{mW\,cm^{-2}}$), the phase-transition temperature of the sample with *cis* form of Azo-F, compared to that with the *trans* form, shifted to a lower temperature without the attenuation of the permittivity. The difference in the phase-transition temperature between the samples with *trans* and *cis* states of Azo-F gradually increased when [Azo-F] was increased (Fig. 2d). Therefore, if we choose an appropriate temperature at which the *trans* and *cis* states induce N$_F$ and M phases, respectively, the dielectric permittivity of the LC blend could be tuned with light irradiation at a constant temperature. To optimise [Azo-F] for this purpose, we evaluated the tunable permittivity range, ($\varepsilon'_{max} - \varepsilon'_{min}$), where $\varepsilon'_{max}$ and $\varepsilon'_{min}$ are the maximum permittivity in the *trans* state and the minimum permittivity in the *cis* state, respectively, at a fixed temperature (Fig. 2b). This range is ~$10^4$ for [Azo-F] of 1–4 wt%, but slightly lower for higher [Azo-F]. We also calculated the relative tunability, [($\varepsilon'_{max} - \varepsilon'_{min}$)/$\varepsilon'_{max}$] to quantify this tendency. All the LC blends showed a high relative tunability of more than 98%, with the LC blend with 4 wt% Azo-F exhibiting the highest value of 99.5% (Fig. 2c). Based on these results, we selected the LC blend with 4 wt% Azo-F as the optimal sample. To confirm the photoinduced M–N$_F$ phase transition, we first observed its polarised optical microscopy images before and after GL irradiation at 52 °C. As expected, the polarised optical microscopic textures in the *trans* and *cis* states correspond to those of the N$_F$ and M phases, respectively (Fig. 2e). On the other hand, no photoinduced M–N$_F$ phase transition occurs at the lower or higher temperature region. For example, at 45 °C (the N$_F$ phase) and 60 °C (the M phase), the light irradiation did not change the polarised optical microscopic textures. To further support the photoinduced M–N$_F$ phase transition, we performed

X-ray diffraction (XRD) measurements (Fig. 2f–i). The two-dimensional (2D) XRD image of the LC blend with 4 wt% Azo-F under a magnetic field (~0.5 T) at 60 °C (the M phase) exhibited a diffraction peak at $q = 0.28\,\mathrm{\AA^{-1}}$ (Fig. 2f), which corresponds to the molecular length of DIO (~2.2 nm). When the temperature was decreased to 52 °C, the peak became stronger and sharper, while its position was maintained, suggesting the induction of the M to N$_F$ phase transition (Fig. 2g). After GL irradiation at 52 °C, the 2D XRD profile was similar to that of the M phase (Fig. 2h), while subsequent BL irradiation returned it to the original N$_F$ phase (Fig. 2i).

**Phototunable dielectric properties of the LC blend**. Figure 3a, b displays the dielectric spectra (relative dielectric permittivity, $\varepsilon'$; dielectric loss, $\varepsilon''$) as a function of frequency during alternate GL/BL irradiation at 50 °C. In the *trans* state of Azo-F, the LC blend ([Azo-F] = 4 wt%) maintained a constant dielectric permittivity ($\varepsilon' = $~18,000) below the frequency of 1 kHz. During GL irradiation (525 nm; $I = 180\,\mathrm{mW\,cm^{-2}}$) for 30 s, the dielectric permittivity decreased gradually and reached ~200 ($f = 1\,\mathrm{kHz}$) in the photostationary state. Simultaneously, a significant reduction in the corresponding dielectric loss was observed. Subsequent BL irradiation (415 nm; $I = 7.0\,\mathrm{mW\,cm^{-2}}$) for 30 s restored the dielectric spectral properties. The response time of the photoisomerization depended on the light intensity (Supplementary Fig. 12). Importantly, this process was completely reversible, and no material degradation was observed even after approximately 100 cycles of alternating GL/BL irradiation (Fig. 3c). Furthermore, the dielectric permittivity in the *cis* state ($\varepsilon' = $~200) was maintained for a long time (>10 h) at 50 °C owing to the long half-life of Azo-F (Supplementary Fig. 13). These results confirm that our LC blend exhibits large phototunability of dielectric permittivity (~$200 < \varepsilon' < $~18,000), rapid photoresponsivity (photoresponse time < 30 s), good fatigue resistance (~100 cycles), and high thermal stability in the *cis* state (>10 h), all of which are essential factors for their practical application. As expected, the dielectric spectra in the *cis* state (52 °C; M phase) are in good agreement with those in the *trans* state at a higher temperature (60 °C; M phase) (Supplementary Fig. 14). When the driving temperature was not in the appropriate range (for example, at 45 or 60 °C), the dielectric spectra in the *trans* and *cis* states were almost identical to each other, and large photoinduced changes in permittivity could not be observed (Supplementary Fig. 15).

As dielectric permittivity is a fundamental parameter that dominates various physical phenomena and properties of materials, many researchers have attempted to tune its value using external stimuli such as photo[40–44], thermo[45–51], electro[52,53], mechano[54], and chemo stimuli[55,56]. The tunable permittivity range, ($\varepsilon'_{max} - \varepsilon'_{min}$) and relative tunability, [($\varepsilon'_{max} - \varepsilon'_{min}$)/$\varepsilon'_{max}$] of our LC blend ([Azo-F] = 4 wt%) and other reported materials, including organic, inorganic and organic/inorganic hybrid materials, are summarised in Fig. 3d (also see Supplementary Table 1). As shown, our LC blend shows the highest relative tunability (99.5%) among the reported materials (also see Supplementary Fig. 16), and its tunable permittivity range ($\varepsilon' = $~200–18,000) is the widest among the permittivity ranges reported for organic and organic/inorganic materials. Importantly, tunability (range and extent) of our LC blend is far beyond those of the previously reported phototunable materials[40–44]. It is worth noting that our LC blend is fluid, whereas most of the reported materials are solids. Because fluidic materials generally provide more flexibility, processability, and scalability than solid materials, our LC blend is expected to have various applications.

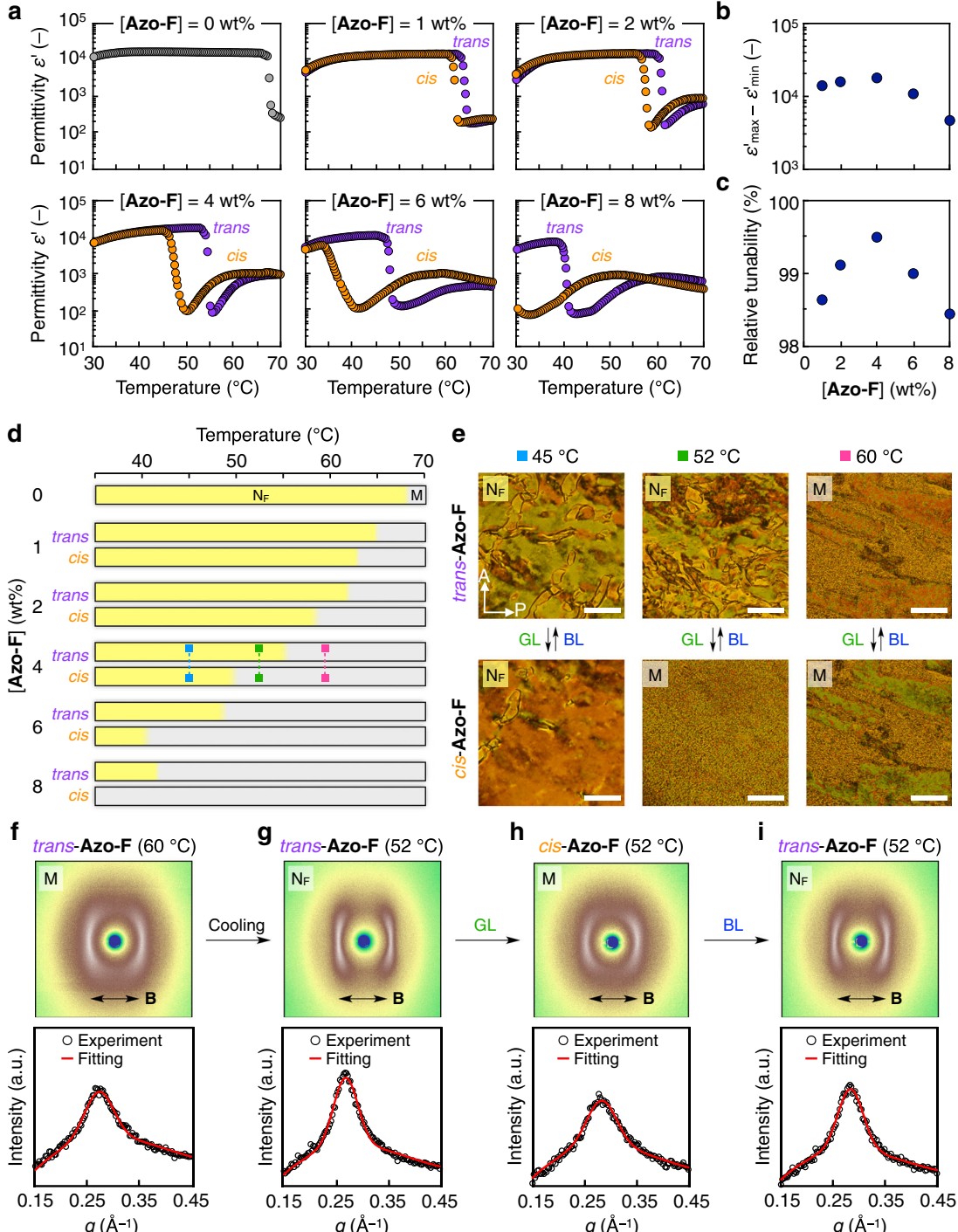

**Fig. 2 Photoinduced transition between mesophase (M) and ferronematic phase (N$_F$). a** Temperature-dependent dielectric permittivity of liquid-crystalline (LC) blends ([Azo-F] = 0–8 wt%) in *cis* (orange plots) and *trans* (purple plots) states. The *cis* and *trans* states were realised by green light (GL; 525 nm) and blue light (BL; 415 nm) irradiation, respectively. **b, c** Phototunable permittivity range, ($\varepsilon'_{max} - \varepsilon'_{min}$); **b** and relative tunability [($\varepsilon'_{max} - \varepsilon'_{min}$)/$\varepsilon'_{max}$; **c** of the LC blends ([Azo-F] = 1–8 wt%), where $\varepsilon'_{max}$ and $\varepsilon'_{min}$ are the maximum permittivity in the *trans* state and the minimum permittivity in the *cis* state, respectively, at a fixed temperature. **d** Temperature-dependent phase behaviours of the LC blends ([Azo-F] = 0–8 wt%) in *cis* and *trans* states. **e** Polarised optical microscopy images under crossed Nicols of an LC blend ([Azo-F] = 4 wt%) in *cis* (lower) and *trans* (upper) states after passing through a long path filter ($\lambda$ > 550 nm) at a constant temperature [left: 45 °C (cyan symbols in (**d**)); middle: 52 °C (green symbols in (**d**)); right: 60 °C (magenta symbols in (**d**))]. Scale: 100 μm. **f–i** Two-dimensional (2D; upper) and one-dimensional (1D; lower) X-ray diffraction profiles of an LC blend ([Azo-F] = 4 wt%) under a magnetic field (**b**; ~0.5 T): in a *trans* state (60 °C; **f**), in a *trans* state (52 °C; **g**), in a *cis* state after GL irradiation (52 °C; **h**), and in a *trans* state after subsequent BL irradiation (52 °C; **i**). Black double arrows indicate the direction of an applied magnetic field. Red lines in the 1D profiles are the best fits.

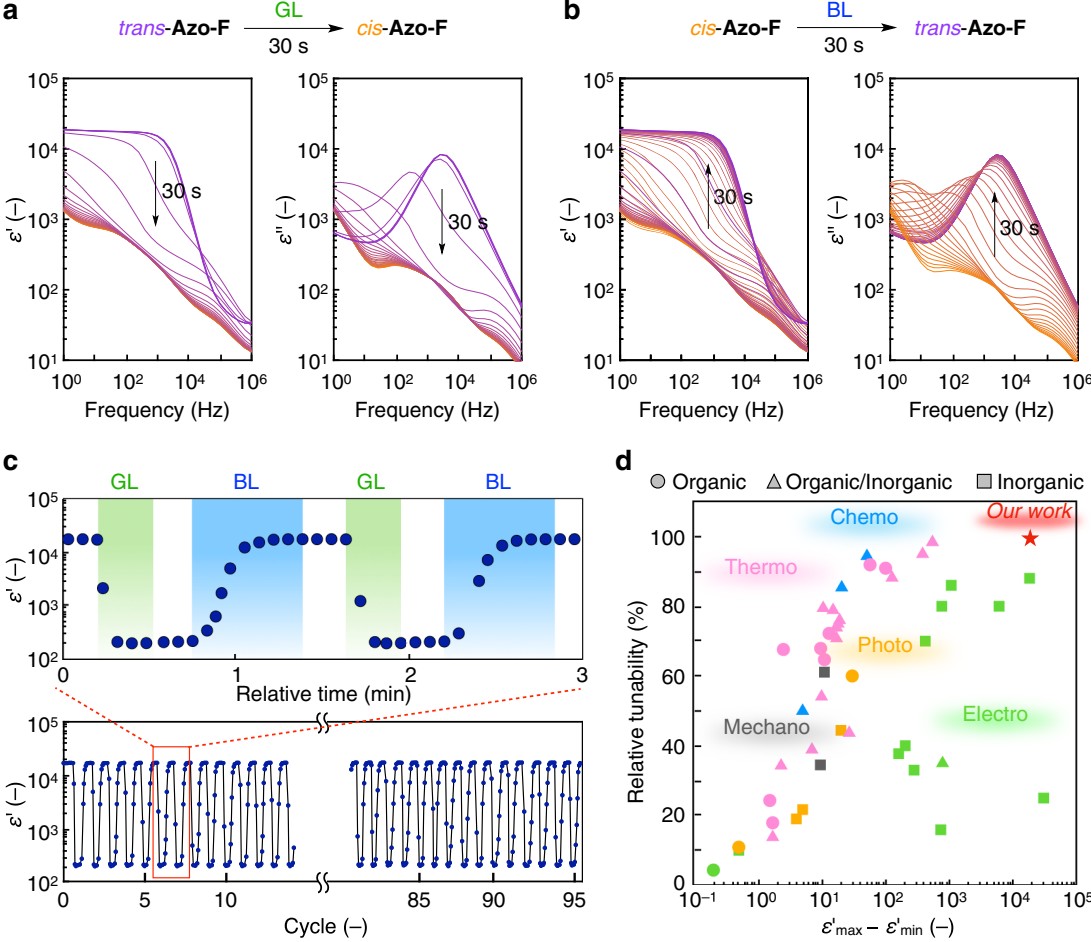

**Fig. 3 Ultralarge-range phototunability of dielectric permittivity. a**, **b** Dielectric spectra [left: relative dielectric permittivity ($\varepsilon'$); right: dielectric loss ($\varepsilon''$)] of a liquid-crystalline (LC) blend ([Azo-F] = 4 wt%) as a function of frequency ($f = 1$–$10^6$ Hz) during green light (GL) (**a**) and blue light (BL) (**b**) irradiation for 30 s at 50 °C. **c** Dielectric permittivity changes ($f = 1$ kHz; 50 °C) during *cis/trans* isomerisation cycles (lower) and their enlarged profiles as a function of relative time (upper). Irradiation times for GL and BL were 20 and 40 s, respectively. **d** Stimuli-responsive tunability of dielectric permittivity of reported materials based on organic (circle), inorganic (square), and organic/inorganic (triangle) components. Stimuli-responsive permittivity range ($\varepsilon'_{max} - \varepsilon'_{min}$) and relative tunability [($\varepsilon'_{max} - \varepsilon'_{min})/\varepsilon'_{max}$] are shown. Photo-, thermo-, electro-, mechano- and chemo-responsive materials are represented by yellow, magenta, green, grey and cyan symbols, respectively.

**Construction and application of a photovariable capacitor.**
Finally, to demonstrate the practical applicability of our LC blend, we constructed a photovariable capacitor using it. As the capacitance of this capacitor is proportional to its dielectric permittivity[57], a substantial change in capacitance is expected under photoirradiation. First, we sandwiched the LC blend ([Azo-F] = 4 wt%) between two glass substrates coated with a transparent indium tin oxide (ITO) electrode with a gap of 17.6 μm to fabricate a photovariable capacitor. Then, we incorporated the resultant capacitor in an electronic circuit with an electric audio oscillator (Fig. 4a, Supplementary Fig. 17, and Supplementary Note 4). We could readily note the change in capacitance by hearing the output sound frequency from a speaker because the sound frequency is inversely proportional to the capacitance of the capacitor. Indeed, alternate BL/GL irradiation for 10 s followed by 5 s of holding in the dark-induced reversible and substantial changes in the sound frequency (Supplementary Fig. 18 and Supplementary Audio 1). The short-time Fourier transform of the data afforded a sound spectrum, as shown in Fig. 4d and Supplementary Fig. 19. After GL irradiation for 10 s, a primary peak was observed at a frequency of ~8.5 kHz (Fig. 4c). During BL irradiation for 10 s, the primary peak

gradually shifted to the lower frequency region (Fig. 4d). Although the primary peak could not be detected after BL irradiation because of the resolution limit of the microphone (~1 kHz), it was estimated to be ~100 Hz using a sound spectrum (Fig. 4b). When the BL irradiation was stopped, the sound frequency was maintained for ~5 min (Supplementary Fig. 20). These results demonstrate that the photovariable capacitor composed of our LC blend realises a substantive photoresponsive change in the output sound frequency in the range of ~100 to 8500 Hz, corresponding to a capacitance change in the range of ~360–4 nF, respectively.

In conclusion, we developed a phototunable fluidic material composed of a ferronematogen (DIO) and a newly designed azobenzene-tethered phototrigger (Azo-F, doping concentration: 4 wt%) to realise a large and reversible change in dielectric permittivity (~200 < $\varepsilon'$ < ~18,000) upon irradiation with blue and green wavelengths. This material exhibits a rapid photoresponse (response time: <30 s), good fatigue resistance (~100 cycles), high thermal stability in the *cis* state (>10 h), and the highest relative tunability of the dielectric permittivity (99.5%) ever reported for materials with stimuli-responsive dielectric permittivity. To demonstrate its applicability, the fluidic material was used to

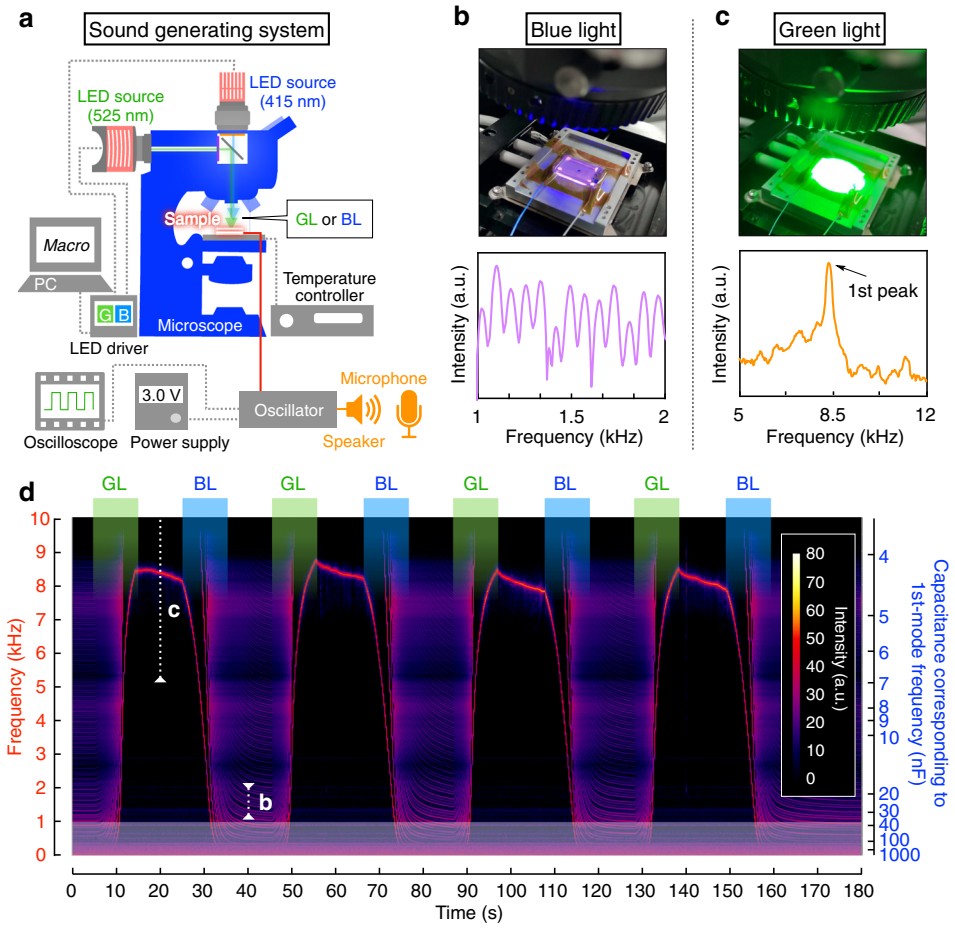

**Fig. 4 Demonstration of a photovariable capacitor operation. a** Schematic illustration of a sound-generating system with a photovariable capacitor consisting of a liquid-crystalline (LC) blend ([Azo-F] = 4 wt%). Sounds generated from an electric speaker were recorded using a microphone. **b,c** Real photoimages (upper) of a photovariable capacitor during blue light (BL) (**b**) and green light (GL) (**c**) irradiation and sound spectra (lower) obtained after BL (**b**) and GL (**c**) irradiation for 10 s. **d** Sound spectrogram generated by short-time Fourier transform of the audio recorded during alternate GL/BL irradiation for 10 s (Supplementary Audio 1). White dotted lines in the spectrogram represent the sound spectra in (**b**) and (**c**). White blurry regions ($f < 1$ kHz) denotes the resolution limit of a microphone.

construct a photovariable capacitor that allowed the reversible modulation of the sound frequency over a wide range ($100 < f < 8500$ Hz) in a remote manner. Such fluidic materials with large and reversible phototunability of dielectric permittivity can be used as a solvent or constituent of soft materials and devices to realise phenomenal advances in fundamental and applied sciences.

## Methods

**Fabrication methods for LC cells.** Sandwich-type electrical cell (17.8 and 21.4 μm-thickness): Partially ITO-coated glass plates (EHC model D-type, electrode area: 50 mm²) were silanized with a silane coupling reagent (octadecyltriethoxysilane, TCI) at 120 °C for 2 h, and then were rinsed with EtOH and ultrapure water. The two resulting glass plates were fixed with drops of UV-curable glue using polymeric beads (micropearl, SEKISUI) as a spacer. The cell gap was estimated by the capacitance of the empty cell.

**Conditions of LC samples.** Preparation of LC blends: The photo-responsive LC blends composed of DIO and Azo-F were prepared by mechanically stirring with a teflon-coated magnetic chip in an aluminium crucible (40 μL, Mettler).

For the experiment without a magnetic field: The LC blend was injected by capillary action into an LC cell (17.8 μm thickness).

For the experiment with a magnetic field: The LC blend was injected by capillary action into an LC cell (21.4 μm thickness).

For the UV–vis study: The LC blend was injected by capillary action into a sandwich-type quartz cell without spacers.

**UV–vis spectroscopy.** UV–vis spectra changes were recorded upon light irradiation at different time intervals, for which the alternating GL and BL irradiation sequence was controlled by a software package for DC2200 LED Driver (Thorlabs) automated with a mouse-action recorder (HiMacroEx, https://fefnirm.web.fc2.com/soft/himacroex/). During recording spectra, a solution of the photo-triggers (Azo-H, Azo-Me and Azo-F) in organic solvent was stirred with a teflon-coated magnetic chip in a quartz cuvette (1 cm × 1 cm, 800 rpm). For UV–vis spectra of the LC blend, [Azo-F], we used a temperature controller (monoOne-120, THREE HIGH) and a handmade hot stage.

**Polarised optical microscopy and dielectric spectroscopy.** As shown in Fig. 4a, the green and blue LED sources were equipped with the polarised optical microscope. For GL irradiation (525 nm), light from the LED source was passed inside of the microscope via a silver mirror, reaching the LC sample mounted on the hot stage (HCS402+mk2000 equipped with LN2-PACD2, INSTEC). In the case of BL irradiation (415 nm), the silver mirror was removed and shined a light on the LC sample directly. Unless otherwise noted, polarised optical microscopy and dielectric spectroscopy were performed simultaneously under this irradiation setup. Polarised optical microscopy and dielectric spectroscopy under a magnetic field were carried out using room temperature bore superconducting magnet systems (9 Tesla, Cryogenic). The measurement system is shown in Supplementary Fig. 7a, b. For Polarised optical microscopy under a magnetic field, we used a handmade hot stage (ITO glass heater); measurement temperature was controlled using a regulated DC power supply (PA18-6A, Kenwood) and monitored via a handmade temperature monitor. For dielectric spectroscopy under a magnetic field, a LED driver (DC2200, Thorlabs) and a white LED (MCWHLP1, Thorlabs) as a backlight and a camera (Powerpack, Basler) for microscopy were used.

**X-ray diffraction (XRD) measurements.** Samples introduced in a glass capillary (1.5 mm in diameter) were measured under a magnetic field at a constant

temperature using a temperature controller with high temperature-resistance neodymium magnets (~0.5 T, MISUMI) fixed in an X-ray diffractometer (NANOPIX, Rigaku). The scattering vector $q$ ($q = 4\pi\sin\theta/\lambda$; $2\theta$ and $\lambda$ = scattering angle and wavelength of an incident X-ray beam [1.54 Å], respectively) and position of an incident X-ray beam on the detector were calibrated using several orders of layer diffractions from silver behenate ($d = 58.380$ Å). The sample-to-detector distance was 71.5283 mm, where acquired scattering 2D images were integrated along the Debye–Scherrer ring by using software (Igor Pro with Nika-plugin), affording the corresponding one-dimensional profiles.

**Demonstration of photo variable capacitor.** Alternating GL and BL light irradiation was repeated, similarly to the UV–vis measurement as described above. An electro audio oscillator system is shown in Fig. 4a. The used oscillator circuit is shown in Supplementary Fig. 17a. In this system, the oscillator was driven using a regulated DC power supply (PA18-6A, Kenwood) and an output-wave form via the oscillator was monitored using an oscilloscope (DSOX2004A, KEYSIGHT). A generated sound from the oscillator via a speaker (Sonitron) was recorded using a microphone. Audio spectra were obtained by short-time Fourier transform (STFT) of the recorded audio data (WAV) using free software, Praat (https://www.fon.hum.uva.nl/praat/) and SoX (http://sox.sourceforge.net). For STFT, we adopted Hamming window as a window function.

**Reporting summary.** Further information on research design is available in the Nature Research Reporting Summary linked to this article.

## Data availability

The authors declare that the data supporting the findings of this study are available within the paper and its supplementary information files. All other information is available from the corresponding authors upon reasonable request.

## Code availability

No customised code was used in this study.

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

## Acknowledgements

We are grateful to Dr. Y. Ishida (RIKEN, CEMS) for allowing us to use a NANOPIX 3.5 m system (Rigaku) and a 9-T superconducting magnet (Cryogenic). We would like thank Dr. D. Okada (RIKEN, CEMS) for preparing a silver mirror coated with SiO$_2$ and Mr. H. Toshima (Kyushu University, IMCE) for the mass spectrometry. This work was partially supported by JSPS KAKENHI Grant Nos. JP19K15438 (H.N.) and JP21H01801 (F.A.), Incentive Research Projects in RIKEN (No. 100689; H.N.), JST PRESTO (Grant No. JPMJPR20A6; K.S.), and JST CREST (Grant Number JPMJCR17N1; F.A.). K.S. acknowledges the Kurita Water and Environment Foundation (KWEF, Japan) and the RIKEN Special Postdoctoral Researcher Programme.

## Author contributions

H.N. conceived the project and designed the experiments. K.S. and F.A. co-designed the experiments. H.N. performed all the experiments. K.S. supported XRD measurements. F.A. constructed an electric audio oscillator. H.N., K.S., and F.A. analysed data and discussed the results. H.N., K.S. and F.A wrote the paper and approved the final paper.

## Competing interests

The authors declare no competing interests.
