## [Peer review file · Nature Communications]

REVIEWER COMMENTS

Reviewer #1 (Remarks to the Author):

In the manuscript »Anisotropic fluid with phototunable dielectric permittivity«, the authors report on the design, properties, and applicability of a newly designed mixture that exhibit phototunable phase transition between nematic and the ferroelectric nematic phase. In the presented work, a new photo trigger molecule was designed and synthesized, which was subsequently mixed with the material DIO. In DIO, the phase transition to the ferroelectric nematic phase is characterized by a huge increase in dielectric permittivity. The authors showed that in the mixtures the phase transition between the nematic and ferroelectric nematic phase can be reversibly induced by illumination with green/blue light. Consequently, also the permittivity of the material either strongly increases or decreases depending on the wavelength of the light used for the illumination. The photoinduced transition was studied by polarized optical microscopy, dielectric spectroscopy, X-ray diffraction, and DSC measurements. The results showed that the material exhibits large phototunability of dielectric permittivity, relatively fast photoresponsivity, good fatigue resistance, and high thermal stability of the cis state of the photo-trigger molecules. The applicability of the designed material was demonstrated by the construction of a photovisible capacitor that allowed the reversible modulation of the sound frequency over a wide range. The design and properties of ferroelectric nematic crystals are currently the hottest topics in liquid crystalline research. In my opinion, this work presents a very important and original contribution not only to the field of liquid crystals but to the wider field of soft matter. The paper is also well organized and very clearly written, and I strongly recommend it for publication in Nature Communications.

Some minor questions

- 1) Does the mixing of DIO with the photo-trigger suppress the crystallization of the DIO?**
- 2) Does the photoresponse time depend on the intensity of the illumination?**

Reviewer #2 (Remarks to the Author):

The manuscript reports an intriguing photo-responsive dielectric permittivity with an unprecedented change from ~ 200 to ~ 18000 through a liquid crystal system. The key LC material DIO undergoes a phase transition from mesophase M to ferronematic phase NF during the temperature cooling. The NF phase shows a gigantic dielectric permittivity (>10000) while the M phase shows small dielectric permittivity (only ~ 200). By doping the well-designed photo-responsive materials based on azobenzene, NF-M phase transition is enabled by light irradiation and accompanied by an enormous change in dielectric permittivity. The azo-F in the LC system facilitates rapid and reversible photoisomerization, good fatigue resistance and high thermal stability. Base on the difference in the NF and M phase temperature range corresponding to the trans and cis states of the doped azo-F material, the materials system achieves photo-induced phase transition at a specific temperature. The author also demonstrated a photovisible capacitor prototype, allowing the modulation of the sound frequency. Overall, this work is very interesting and significant, and the experimental results are solid. I would recommend this manuscript to be accepted for publication after making some minor revisions.

- 1. The description of N phase and M phase looks not very clear.**
- 2. Usually the azobenzene materials are changed from trans to cis by short and wavelength light (UV or blue), why the azo-F here can be transferred from trans to cis by green light and cis to trans by blue light. What might the deep mechanism of this phenomenon be?**
- 3. In SI Fig. 9, it looks like there is only trans state corresponding to POMs and DSC data of**

phase transition, how about the results of cis state?

4. When the temperature increases, the dielectric corresponding to the trans state decreases rapidly at the phase transition temperature point, while the cis state decreases slowly, what is the reason?

5. The authors may consider citing the references related to the diverse control of materials properties using light stimulus (e.g., *Acc. Chem. Res.* 2014, 47, 3184; *Angew. Chem. Int. Ed.* 2016, 55, 2994; *Nature* 2016, 531, 352) and light-driven liquid crystal phase change (e.g., *Chem. Rev.* 2016, 116, 15089; *Adv. Mater.* 2018, 30, 1800237; *Mater. Today* 2017, 20, 230).

Reviewer #3 (Remarks to the Author):

The authors have designed a light-controllable liquid crystal (LC) material with a very large and reversible change in dielectric permittivity. To this end, the DIO liquid crystal with the ferronematic phase was doped with a newly synthesized fluorinated azobenzene-tethered molecular photoswitch (Azo-F). Both the trans-cis and cis-trans isomerizations of azo-based molecules occurs upon visible light irradiation at two different wavelengths, while the thermal stability of the cis state is quite high. Light-induced isomerization leads to a phase transition from the ferronematic phase to the mesophase and back, providing a photo-triggered change in dielectric permittivity. The switching times between the two liquid crystal states are short and the reversibility is fatigueless over a large number of switching. The capacitor, composed of the developed light-responsive LC mixture, realizes photo-reversible modulation of the output sound frequency in a wide range from hundreds to thousands of Hz. Overall, the results are well presented and interesting.

There are, however, a few comments which I'd like the authors to address:

1. Was the photostationary state reached after the irradiation time indicated in Fig.1b,e,h?
2. Both isomerization reactions are induced by visible light. Have the experiments been carried out in dark room conditions?
3. The half-life time of cis-Azo-F in DMSO at room temperature is 128 days. The stability of dielectric permittivity of the LC material with cis-Azo-F at 52 degrees is 10 h only. Why are the timescales so different?
4. Pure LC was doped with Azo-F up to 8 wt%, which allowed to obtain the M-Nf phase transition at about 30 degrees. Why were not higher concentrations tested? Despite a possible further decrease in the relative tunability, the advantage of using the LC material at room temperature seems important.

Minor comments:

5. Fig.1h: was the green light illumination performed at 530 nm or 525 nm? Please check.
6. What does N' mean in Fig.2e? The M phase, perhaps? The same question for the caption to Supplementary Fig.9.
7. Page 7, the phrase "Thus, the n-n* bands of the cis- and trans- isomers of Azo-Me were successfully separated..." I think this is about Azo-F, i.e. it should be "Thus, the n-n* bands of the cis- and trans- isomers of Azo-F were successfully separated...". Please check.

Point-by-Point Responses to Reviewers' Comments

First of all, the authors would like to express our sincere gratitude to all the three reviewers for carefully reading our manuscript and giving us their invaluable comments and feedbacks. We truly appreciate their supportive and encouraging comments on our work.

To Reviewer #1

(Reviewer's Comment) In the manuscript »Anisotropic fluid with phototunable dielectric permittivity«, the authors report on the design, properties, and applicability of a newly designed mixture that exhibit phototunable phase transition between nematic and the ferroelectric nematic phase. In the presented work, a new photo trigger molecule was designed and synthesized, which was subsequently mixed with the material DIO. In DIO, the phase transition to the ferroelectric nematic phase is characterized by a huge increase in dielectric permittivity. The authors showed that in the mixtures the phase transition between the nematic and ferroelectric nematic phase can be reversibly induced by illumination with green/blue light. Consequently, also the permittivity of the material either strongly increases or decreases depending on the wavelength of the light used for the illumination. The photoinduced transition was studied by polarized optical microscopy, dielectric spectroscopy, X-ray diffraction, and DSC measurements. The results showed that the material exhibits large phototunability of dielectric permittivity, relatively fast photoresponsivity, good fatigue resistance, and high thermal stability of the cis state of the photo-trigger molecules. The applicability of the designed material was demonstrated by the construction of a photovisible capacitor that allowed the reversible modulation of the sound frequency over a wide range. The design and properties of ferroelectric nematic crystals are currently the hottest topics in liquid crystalline research. In my opinion, this work presents a very important and original contribution not only to the field of liquid crystals but to the wider field of soft matter. The paper is also well organized and very clearly written, and I strongly recommend it for publication in Nature Communications.

Some minor questions

Thank you very much for your positive and encouraging evaluation for our manuscript. Please find our point-by-point replies to your questions in the following. We hope these properly answer your concerns and satisfy you.

(Reviewer's Comment 1) Does the mixing of DIO with the photo-trigger suppress the crystallization of the DIO?

=> This is a very good point. If we can realize N_FLC with a wide operating temperature range across RT, it may deliver a great diversity of applications. As you might expect, the doped photo-trigger effectively shifts the phase transition temperatures due to the impurity effect. In fact, we examined the higher concentrations (over 8 wt%), but unfortunately, complete suppression of the crystallization was not obtained in the present case. Actually, we also reported a similar result in our recent work, i.e. addition of a dopant cannot completely suppress the crystallization of **DIO** [Nishikawa, H. et al., *Adv. Mater.*, **33**, 2101305 (2021); Ref. 29]. Mandle et al. are still struggling in blending the RT N_FLC using various N_FLCs, despite their huge efforts on design

and synthesis of new N_FLC molecules [Mandle, R. et al., *Liq. Cryst.* (2021), doi.org/10.1080/02678292.2021.1934740]. The Boulder group also reported alteration of the phase sequence upon mixing of two N_FLCs, **DIO** and RM734 [Chen, X. et al. arXiv:2110.10826v1 (2021); Ref. 33], but it was not extended to RT. While these works don't guarantee the suppression of crystallization by mixing, the use of mixture N_FLCs must be one of the best strategies at present, from the point of view of doping effects to downshift/suppress crystallization with keeping the emergence of the N_F phase. At any rate, the complete suppression of crystallization just by doping or mixing is still a significant challenge.

(Reviewer's Comment 2) Does the photoresponse time depend on the intensity of the illumination?

=> Yes, it does. The response time strongly depends on the light intensity, because in principle the *cis-trans* isomerization ratio changes with total input light energy. In the present paper, we have demonstrated the photoresponse, quickest with the maximum light intensity available in our apparatus (Fig. 3c, $I_{GL} = 180 \text{ mW cm}^{-2}$ at 525 nm and $I_{BL} = 7.0 \text{ mW cm}^{-2}$ at 415 nm). But, of course, faster photoresponse will be obtained, if any higher-power light source is available instead of ours. For this technical reason of the present setup, as for the light intensity dependence, we can observe only lowering of response time by reducing the irradiation intensity (see the figure below for 10-90 response times). In this case, the shape of the photoresponse curve alters from rectangular to more sinusoidal (newly-added Supplementary Figure 12), because of the slower response under the weak light irradiation. Since we noticed that the response upon varying irradiation intensity was not mentioned in the manuscript, we added a short description of this to the main text on page 11, line 1–2 and added the corresponding data in Supplementary Fig. 12.

To Reviewer #2

(Reviewer's Comment) The manuscript reports an intriguing photo-responsive dielectric permittivity with an unprecedented change from ~ 200 to ~ 18000 through a liquid crystal system. The key LC material DIO undergoes a phase transition from mesophase M to ferronematic phase NF during the temperature cooling. The NF phase shows a gigantic dielectric permittivity (>10000) while the M phase shows small dielectric permittivity (only ~ 200). By doping the well-designed photo-responsive materials based on azobenzene, NF-M phase transition is enabled by light irradiation and accompanied by an enormous change in dielectric permittivity. The azo-F in the LC system facilitates rapid and reversible photoisomerization, good fatigue resistance and high thermal stability. Base on the difference in the NF and M phase temperature range corresponding to the trans and cis states of the doped azo-F material, the materials system achieves photo-induced phase transition at a specific temperature. The author also demonstrated a photovisible capacitor prototype, allowing the modulation of the sound frequency. Overall, this work is very interesting and significant, and the experimental results are solid. I would recommend this manuscript to be accepted for publication after making some minor revisions.

=> Thank you very much for your positive and encouraging evaluation for our manuscript. Please find our point-by-point replies to your questions/comments in the following. We hope these properly answer your concerns and satisfy you.

(Reviewer's Comment 1) The description of N phase and M phase looks not very clear.

=> We should admit that the N and M phases were not well explained. In response to this constructive comment, we revised the corresponding part in the main text in Page 4, line 3–4 as follows:

“Importantly, **DIO** exhibits the N–N_F phase transition across the additional mesophase (M) which is presumably having a local antiferroelectric nematic order (Refs. 32,33).”

(Reviewer's Comment 2) Usually the azobenzene materials are changed from trans to cis by short and wavelength light (UV or blue), why the azo-F here can be transferred from trans to cis by green light and cis to trans by blue light. What might the deep mechanism of this phenomenon be?

=> The commonly-known azobenzenes have two distinct absorption bands around 365 nm (UV, $\pi \rightarrow \pi^*$ transition) and 450 nm (blue, $n \rightarrow \pi^*$ transition) in the ground trans-isomer state, and the former is usually irradiated to induce the trans-to-cis photoisomerization. As for the back reaction process, i.e., the cis-to-trans photoisomerization, the $n \rightarrow \pi^*$ transition of the cis-isomer is utilized with the blue light irradiation. However, this means that the absorption band at the blue region is in fact the superimposed $n \rightarrow \pi^*$ transition bands of trans- and cis-isomers. Thus, if the $n \rightarrow \pi^*$ transition bands of the trans- and cis-isomers can be effectively separated in the visible region, we are able to utilize the trans $n \rightarrow \pi^*$ transition to induce the trans-to-cis photoisomerization and don't have to rely on the harmful UV light. In the case of Azo-F, the $n \rightarrow \pi^*$ band of

the *cis*-isomer is blue-shifted due to the lower energy of the n-orbital of the *cis*-isomer (because the fluorine substituents in the ortho-positions, i.e. nearby σ -electron-withdrawing groups, effectively reduce the n-electron density), resulting in the effective separation of the $n \rightarrow \pi^*$ absorption bands as shown in Fig. 1, and then the green light (above 500 nm) becomes accessible to the *trans*-to-*cis* isomerization through the $n \rightarrow \pi^*$ transition of the *trans*-isomer. Bleger et al. discussed such a band separation process based on the molecular orbital (MO) theory in Ref. 36 in the main text.

In response to this comment, we added some discussions on this point as Supplementary Note 2.

(Reviewer's Comment 3) In SI Fig. 9, it looks like there is only *trans* state corresponding to POMs and DSC data of phase transition, how about the results of *cis* state?

=> In response to this suggestion, we conducted additional experiments on POM observation of the *cis* state in various [*cis*-**Azo-F**] of 0–8 wt% (Supplementary Fig. 10a). As for the DSC in the *cis*-state, since there is no accessible photo-DSC machine around us, it was difficult to examine the calorimetric property under light irradiation. Instead, we measured the DSC traces for a post-irradiated sample (Supplementary Fig. 11). Fortunately, the relaxation time of the *cis*-state of the present [**Azo-F**] is relatively long, and hence it was possible to complete the DSC scans within a sufficiently short time after green light irradiation. Because the data looks nice and provides some additional useful information on the thermo-dynamic behavior of the system such as the thermal-back property, it was newly added as Supplementary Fig. 11 together with some descriptions. We appreciate this valuable comment by the reviewer.

(Reviewer's Comment 4) When the temperature increases, the dielectric corresponding to the *trans* state decreases rapidly at the phase transition temperature point, while the *cis* state decreases slowly, what is the reason?

=> First, we would like to note that the data in Fig. 2a are the comparison of the effect of blue or green light on the permittivity data recorded during cooling. As you mentioned, as for the low azo-content samples ([**Azo-F**] \leq 4 wt%), the dielectric permittivity values of both *trans*- and *cis*-isomers rapidly increased at the phase transition with reducing temperature, i.e., the dielectric curves are just shifted toward lower temperature. On the other hand, in the case of high azo-content samples ([**Azo-F**] \geq 6 wt%), the dielectric permittivity of *cis*-isomer gradually increased as you point out. In this case, because of the strong absorption due to the high concentration of **Azo-F**, light didn't penetrate deep into the sample and unreacted *trans*-isomer is remaining even under light irradiation. For this reason, there is certain inhomogeneity in the sample. Indeed, this photoinduced phase transition in the higher azo-content samples seems to start from the surface. So, by using higher intensity green light, this increase may become steeper.

(Reviewer's Comment 5) The authors may consider citing the references related to the diverse control of materials properties using light stimulus (e.g., Acc. Chem. Res. 2014, 47, 3184; Angew. Chem. Int. Ed. 2016,

55, 2994; Nature 2016, 531, 352) and light-driven liquid crystal phase change (e.g., Chem. Rev. 2016, 116, 15089; Adv. Mater. 2018, 30, 1800237; Mater. Today 2017, 20, 230).

=> Thank you for suggesting including these relevant papers we missed. These papers are cited as Refs. 9–14 in the revised version of the manuscript.

To Reviewer #3

(Reviewer's Comment) The authors have designed a light-controllable liquid crystal (LC) material with a very large and reversible change in dielectric permittivity. To this end, the DIO liquid crystal with the ferronematic phase was doped with a newly synthesized fluorinated azobenzene-tethered molecular photoswitch (Azo-F). Both the trans-cis and cis-trans isomerizations of azo-based molecules occurs upon visible light irradiation at two different wavelengths, while the thermal stability of the cis state is quite high. Light-induced isomerization leads to a phase transition from the ferronematic phase to the mesophase and back, providing a photo-triggered change in dielectric permittivity. The switching times between the two liquid crystal states are short and the reversibility is fatigueless over a large number of switching. The capacitor, composed of the developed light-responsive LC mixture, realizes photo-reversible modulation of the output sound frequency in a wide range from hundreds to thousands of Hz. Overall, the results are well presented and interesting.

=> Thank you very much for your positive and encouraging evaluation for our manuscript. Please find our point-by-point replies to your questions/comments in the following. We hope these properly answer your concerns and satisfy you.

(Reviewer's Comment 1) Was the photostationary state reached after the irradiation time indicated in Fig.1b,e,h?

=> Yes, it was. To answer this question, we analyzed the UV-Vis spectra (Fig. 1b,c,e,f,h,i) and obtained the time-dependent absorbance changes upon light irradiation (Supplementary Fig. 8). As shown in these profiles, they are in the photostationary state after the irradiation times indicated in Fig. 1. We added the related data to Supplementary Fig. 8.

(Reviewer's Comment 2) Both isomerization reactions are induced by visible light. Have the experiments been carried out in dark room conditions?

=> All the experiments (UV-Vis spectroscopy, POM observation, XRD measurement, dielectric spectra and sound generation) were conducted in dark environments. Note that, environmental lights such as from the room lamps were not so strong in our lab, even in the bright state (less than 0.2 mW cm^{-2} at the tabletop). Therefore, we might not need any further special cares to eliminate the side effect from such environmental lights.

(Reviewer's Comment 3) The half-life time of cis-Azo-F in DMSO at room temperature is 128 days. The stability of dielectric permittivity of the LC material with cis-Azo-F at 52 degrees is 10 h only. Why are the timescales so different?

=> Generally, the *cis*-isomer thermally relaxes to the energetically stable *trans*-state in dark. The kinetics of this thermal back *cis*-to-*trans* reaction strongly depends on temperature of system. As known, the thermal back reaction is suppressed at low temperatures, whereas it is promoted at high temperatures. Thus, although the *cis*-Azo-F can persist for a long time (half-life time = 128 days) at RT, the *cis*-Azo-F cannot survive longer at 52°C. Similarly, other azo compounds, *cis*-Azo-H and *cis*-Azo-Me, also exhibits the same tendency. In these cases, the half-times of *cis*-Azo-H (1.1 days) and *cis*-Azo-Me (65 days) are reduced to ~10 min and ~9 hrs, respectively, at about 50°C, as shown in the data below. Of course, this means that we will be able to suppress the thermal back and stabilize the photoswitching property, if the room temperature N_FLC is materialized as mentioned below. For Azo-F, we added a short description of this to the caption of Supplementary Fig. 13.

(Reviewer's Comment 4) Pure LC was doped with Azo-F up to 8 wt%, which allowed to obtain the M-N_F phase transition at about 30 degrees. Why were not higher concentrations tested? Despite a possible further decrease in the relative tunability, the advantage of using the LC material at room temperature seems important.

=> This is a very good point. If we can realize N_FLC with a wide operating temperature range across RT, it may deliver a great diversity of applications. As you might expect, the doped photo-trigger effectively shifts the phase transition temperatures due to the impurity effect. In fact, we examined the higher concentrations (over 8 wt%), but unfortunately, complete suppression of the crystallization was not obtained in the present case. Actually, we also reported a similar result in our recent work, i.e. addition of a dopant cannot completely suppress the crystallization of DIO [Nishikawa, H. et al., *Adv. Mater.*, **33**, 2101305 (2021); Ref. 29]. Mandle et al. are still struggling in blending the RT N_FLC using various N_FLCs, despite their huge efforts on design and synthesis of new N_FLC molecules [Mandle, R. et al., *Liq. Cryst.* (2021), doi.org/10.1080/02678292.2021.1934740]. The Boulder group also reported alteration of the phase sequence upon mixing of two N_FLCs, DIO and RM734 [Chen, X. et al. arXiv:2110.10826v1 (2021); Ref. 33], but it was not extended to RT. While these works don't guarantee the suppression of crystallization by mixing, the use of mixture N_FLCs must be one of the best strategies at present, from the point of view of doping effects

to downshift/suppress crystallization with keeping the emergence of the N_F phase. At any rate, the complete suppression of crystallization just by doping or mixing is still a significant challenge.

(Reviewer's Comment 5) Fig. 1h: was the green light illumination performed at 530 nm or 525 nm? Please check.

=> Thank you for the careful reading of our manuscript. This must be '525 nm'. We corrected Fig. 1h.

(Reviewer's Comment 6) What does N' mean in Fig. 2e? The M phase, perhaps? The same question for the caption to Supplementary Fig. 9.

=> Thanks also for careful checking in details. It must be 'M phase'. We corrected Fig. 2e and accordingly the caption in Supplementary Fig. 10.

7. Page 7, the phrase "Thus, the $n-\pi^*$ bands of the cis- and trans- isomers of Azo-Me were successfully separated..." I think this is about Azo-F, i.e. it should be "Thus, the $n-\pi^*$ bands of the cis- and trans- isomers of Azo-F were successfully separated...". Please check.

=> Sorry for many typos. Again, we would like to express our deepest gratitude to the reviewer for careful reading. It must be '**Azo-F**' as you point out. The statement was corrected. We corrected the main text on page 7, line 9.

List of Revisions

All the revisions are highlighted by yellow in the main text, as well in Supporting Information.

1. We conducted additional experiments on light power dependence of dielectric tunability and added a short description of this to the main text on page 11, line 1–2 and the corresponding data in Supplementary Fig. 12.
2. A short description of the N and M phases in the main text in Page 4, line 3–4.
3. We added some discussions on the mechanism of photoisomerization for **Azo-F** to Supplementary Note 2.
4. Additional POM images of the *cis* state in various [**Azo-F**] to Supplementary Fig.10a.
5. Additional DSC data of the *trans* and *cis* states for [**Azo-F**] (4 wt%) to Supplementary Fig.11.
6. Additional six references, Refs. 9–14 in the revised version of the manuscript.
7. We added the data related to the photostationary state of [**Azo-F**] in solution to Supplementary Fig.8.
8. A short description of the thermal relaxation of azo materials in solution and LC to the caption of Supplementary Fig. 13.
9. The wavelength of the green light illumination was wrong in Fig. 1h in the original manuscript. This was corrected.
10. The nomenclature of the middle mesophase of **DIO** in Fig. 2a was wrong. Thus, we corrected this and the caption in Supplementary Fig.10, accordingly.
11. There was a typo in the main text on page 7, line 9. This was corrected.

REVIEWERS' COMMENTS

Reviewer #1 (Remarks to the Author):

In the revised manuscript, the authors satisfactorily answered the reviewers' comments and, I highly recommend it for publication in Nature Communications.

Reviewer #2 (Remarks to the Author):

I have looked over the authors' point-by-point response and revisions. I think the authors have well addressed the questions raised by the reviewers. I would recommend this revised manuscript to be accepted for publication.

Reviewer #3 (Remarks to the Author):

The authors have carefully revised the manuscript in accordance with the comments made. All questions are clarified, the main text and supplementary materials are enriched with additional data. I would like to recommend this manuscript for publication.

RIKEN Center for Emergent Matter
1-2 Hirosawa, Wako, Saitama 351-0198,
Tel: +81-48-462-1111
Fax: +81-48-467-9599
<http://www.cems.riken.jp/en/>

Point-by-Point Responses to Reviewers' Comments

Dear Reviewers,

The authors would like to express our sincere gratitude to all the three reviewers for carefully reading our manuscript and giving us their invaluable comments and feedbacks. We truly appreciate their supportive and encouraging comments on our work.

Fumito Araoka on behalf of the authors

REVIEWERS' COMMENTS

Reviewer #1:

In the revised manuscript, the authors satisfactorily answered the reviewers' comments and, I highly recommend it for publication in Nature Communications.

Reviewer #2:

I have looked over the authors' point-by-point response and revisions. I think the authors have well addressed the questions raised by the reviewers. I would recommend this revised manuscript to be accepted for publication.

Reviewer #3:

The authors have carefully revised the manuscript in accordance with the comments made. All questions are clarified, the main text and supplementary materials are enriched with additional data. I would like to recommend this manuscript for publication.